# Performance Evaluation of Structural Health Monitoring System Applied to Full-Size Composite Wing Spar via Probability of Detection Techniques

**DOI:** 10.3390/s24165216

**Published:** 2024-08-12

**Authors:** Bernardino Galasso, Monica Ciminello, Gianvito Apuleo, David Bardenstein, Antonio Concilio

**Affiliations:** 1Adaptive Structures Division, The Italian Aerospace Research Centre (CIRA), 81043 Capua, Italy; 2Research Division, Piaggio Aerospace Industries (PAI), 20100 Villanova D’Albenga, Italy; 3Advanced Structural Technologies, Engineering Center, Israel Aerospace Industries (IAI), Ben Gurion International Airport, Tel Aviv 70100, Israel

**Keywords:** probability of detection, structural health monitoring, aeronautic composite structures, de-bonding, FE simulations

## Abstract

Probability of detection (POD) is an acknowledged mean of evaluation for many investigations aiming at detecting some specific property of a subject of interest. For instance, it has had many applications for Non-Destructive Evaluation (NDE), aimed at identifying defects within structural architectures, and can easily be used for structural health monitoring (SHM) systems, meant as a compact and more integrated evolution of the former technology. In this paper, a probability of detection analysis is performed to estimate the reliability of an SHM system, applied to a wing box composite spar for bonding line quality assessment. Such a system is based on distributed fiber optics deployed on the reference component at specific locations for detecting strains; the attained data are then processed by a proprietary algorithm whose capability was already tested and reported in previous works, even at full-scale level. A finite element (FE) model, previously validated by experimental results, is used to simulate the presence of damage areas, whose effect is to modify strain transfer between adjacent parts. Numerical data are used to verify the capability of the SHM system in revealing the presence of the modeled physical discontinuities with respect to a specific set of loads, running along the beam up to cover its complete extension. The POD is then estimated through the analysis of the collected data sets, wide enough to assess the global SHM system performance. The results of this study eventually aim at improving the current strategies adopted for SHM for bonding analysis by identifying the intimate behavior of the system assessed at the date. The activities herein reported have been carried out within the RESUME project.

## 1. Introduction

POD techniques are numerical tools aimed at establishing the reliability of certain procedures in detecting the specific target, based on non-exhaustive investigations. They were initially used as a measure of the capability of NDT methods for structural health assessment during inspection. Recently, POD has been explicitly referred to as a main requirement for the airworthiness of SHM systems, which are required to exhibit high reliability and probability of detection, consistent with standards. Such an approach allows one to interchange the reliability of the structure or structural parts, as adhesive bonds, with the reliability of the SHM system [1,2,3]. Directly derived from NDE applications, POD may be generally defined as a metric aimed at describing the accuracy of a test. For instance, such a statistical method measures the goodness of an inspection procedure in identifying critical defects [4]. In synthesis, POD evaluates the smallest irregularity size and merges quantitative and qualitative parameters. In its publication E2862-12 (then substituted by the versions −18 and −23), ASTM defines POD as the *fraction of nominal discontinuity sizes expected to be found given their existence* [5]. Other studies, however, refer to the POD as the simple probability of detecting a flaw, depending on its size [6]. This is the pattern according to which the study reported in this paper moves along.

In general, different statistical methods are available for estimating POD. Monte Carlo simulation, for instance, has been proposed to derive quantitative data based on nuclear plants’ surveillance. In that case, those features were assigned to simulate many different configurations of the inspection system. As a defect model is assessed, the approach allows for establishing the major deviations assigned to certain monitoring processes [7]. Another original technique has been recently presented, aimed at assessing SHM system reliability, focusing on the quantification of the localization accuracy. The base of the methodology consists of combining several damage scenarios to generate a single POD function describing the designated structural health monitoring system detection capability at a single curve flaw location [8].

The POD is aimed at issuing a quantitative number for defining the reliability of detecting damage. It may be stated that such a tool has allowed a better understanding of NDE procedures and their dependence on operational parameters (as the environment, the used material, the specific application, and so on). In 1997, the NTIAC (Nondestructive Testing Information Analysis Center), published a reference data book to organize the data collected through a number of NDE experiences [9]. A first, a significant overview of the POD to assess NDE reliability was produced in 2001 [10]. There, the author introduced a systematic rationale (or protocol) to quantify the effect of many variables on NDE outcomes, including human factors such as the inspector’s job type and two kinds of comfort perception. Initially developed for the USAF (United States Air Force), the work is general and may easily be exploited for a variety of applications, and even to other industries.

As a natural evolution of NDE, SHM aims at improving the safety and performance of aircraft during service life, while reducing maintenance costs. Such systems find a very wide variety of applications and, in certain cases, almost assume the role of enabling technologies. Specifically thinking to aircraft applications, for instance, the use of fiber optics has certain advantages in SHM systems, being immune to electromagnetic radiation and inert for explosive environments, with the wing box usually being used as a fuel tank [11,12,13]. In addition, unmanned aerial surveillance vehicles may require a high level of safety and reliability, depending on their use and the regions which they fly over. It is not the case that the increase in both military and civil applications has led to the introduction of dedicated regulations, addressing the safety of operations over populated areas [14].

Many NDE-devoted approaches have then been adapted to develop specific metrics for SHM systems evaluation, even though direct applications were not negligible, for many complexities added to those new technologies. In fact, while NDE refers mainly to static architectures being monitored and analyzed, SHM often does interface or is devoted to structures operating in their own environment. Therefore, aspects like the time-dependency of data, or the complexity itself of the investigated architectures, have increased the difficulties of a satisfactory implementation of POD methodologies. A wide review concerning the difficulties of the practical use of suitable metrics for measuring SHM performance and consistency is reported in [15].

SHM techniques have found fertile ground with composite structures, since they may exhibit many defects that can be associated with the very first phase of the manufacturing. One of the most challenging issue is the so-called kissing bond phenomenon, defined as the situation occurring when two surfaces are completely disbanded, or even partially bonded, but directly touching each other in a so-called “dry contact”, or in very close proximity [16]. As a consequence, many NDE ultrasound technologies (like C-Scan) are sometimes not able to detect this kind of defect [17,18].

In this paper, variable damage measures are taken as input features for a non-exhaustive POD analysis to estimate the reliability of an SHM system applied to detect kissing bond-type defects, artificially generated during the manufacturing of a full-size composite spar. This analysis is essential for assessing the actual performance of the structural health monitoring technique, including the sensor network, the algorithm, and the preprocessing data analysis. In fact, the performance verification, which is necessarily applied to a case study in order to maintain a direct link with reality, allows us to identify the point on which the system should be improved based on the actual outcomes. In order to ensure that this investigation is satisfactory, the predictions shall be translated into operational updates and design guidelines. This kind of approach has some degrees of innovation with respect to what can be usually find in the literature, and the authors foresee further improvements in handling this strategy in future works. In what follows, the test article is described with its finite element model representation, including the imposed flaws and the fiber optic sensing system, deployed in the region of interest. Finally, an analysis of the collected data for the POD and the related results are presented and discussed.

## 2. Test Article for Numerical Model Validation

The beam is made of two C-spars and two skin plates, bonded together to form a closed, rectangular cross-section, shown in Figure 1. The spar consists of two unidirectional carbon–epoxy composite plates and two C-section stringers. The plates are bonded to the spar caps, whose width is about 20 mm. The beam has a total length of 1600 mm, a base of 120 mm, and a height of about 90 mm. During the manufacturing process, three artificial damage regions were introduced between the upper skin and one of the stringer caps by deploying Kevlar patches of different lengths and widths between the interfaces of the two structural elements. The test article was manufactured by Israel Aerospace Industries (IAI).

In the FEM representation (realized with the commercial software MSC Nastran 2021), the skins, the spar caps, and the adhesive layer are modeled by HEX solid elements with equivalent 3D orthotropic properties, while spar webs are modeled by 2D QUAD elements.

The three regions are referred to as D5, D2, and D3 in Figure 2. The disbonding widths of D3 and D5 are 20 mm, while D2 is 10 mm wide. In the longitudinal direction (i.e., along the max size), the disbonding lengths are 70, 80, and 40 mm long (Figure 2). These damage zones are modeled by simulating the absence of the adhesive at the interface between the cap and skin *(kissing bond phenomenon)*, by assigning to the elements of the adhesive a negligible Young’s modulus (10^−8^ order of magnitude, just to avoid numerical instability). This value represents a significant reduction in stiffness compared to the inherent properties of the undamaged bonding material. The convergence of such a procedure was tested in [19].

The characteristic mesh length is 8 mm; however, to intercept local strain variation, due to geometrical discontinuity, it was decided to introduce the finest mesh in proximity to the damage zones, as shown in Figure 3.

Monitoring of the structural de-bonding is implemented by using fiber optic-based sensors, devoted to measure local strain. Fiber may be modeled in different ways; a preliminary analysis was specifically carried out to correlate the impact of adopting such FEM methodologies with the achieved results. In detail, three different FEM models were tested, identified as *Segment*, *Merge*, and *Virtual Contact*, for pure classification scopes.

Figure 4 depicts a portion of the model discretized using the *Segment* method. This approach employed a mesh with a 1D element, 5 mm long, with the fiber path approximated to a broken line. The fiber path is deployed over the structure, modeled by brick elements. The connection between the fiber and the plate is achieved through coincident nodes, common to the beam and the brick elements. This method is valuable for rapid preliminary analyses, requiring minimal resources for mesh generation and solution achievement.

The *Merge* methodology employs a finer mesh; in this specific case, 2 mm long elements were used. This approach tries to follow the actual path of the optical fiber along the reference surface, where it has been deployed. Compared to the former method, it implements shorter beam elements, and therefore gives access to more strain evaluation points. In this approach, the segments of the fiber path coincide with the edges of the 3D structure elements along the monitored surface; as before, the fiber and structural nodes are coincident. Figure 5 illustrates the distribution of beams on the panel’s main surface. This meshing strategy offers a more accurate representation of the fiber path, which is particularly relevant for curved portions. Here, the fiber is generally discretized into a finer mesh, compatible with the resolution applied for the analyzed structure. Of course, this method necessitates larger attention during the mesh generation phase and has a larger impact on the full model’s dimension.

As with the mesh generated for the *Merge* approach, the *Virtual Contact* method tries to give a detailed discretization of the fiber path, shown in Figure 6, but with the crucial distinction of avoiding the actual coincidence between the structure and fiber nodes. To simulate their interaction, a surface-to-surface contact is introduced in the way allowed by the FE code (a dedicated contact card shall be generated). In this case, the fiber cross-section is simulated as an equivalent square section (instead of referring to a circular section) of the same area. In spite of an apparent simplification, this approach permits a better simulation of the contact area between the fiber and the surface through the adhesive layer.

The three methods are plotted in Figure 7 for comparison. While the chosen method (Segment) utilizes the simplest representation of the fiber’s physical behavior, it is still applicable for the SHM target since the algorithm implemented does not take into consideration the absolute strain but does for relative variations (as explained in Section 3). The numerical strain estimation demonstrated good agreement with the experimental strain measurements, [19,20,21].

The experimental–numerical correlation was established through a three-point bending test conducted on a dedicated test rig. The test setup included two end supports for the spar installation and a linear actuator capable of applying up to 10 kN vertical load at the beam center. In the validation tests, such a load was limited to 5 kN to prevent excessive deformation of the structural element and subsequent damage. A cursor, connected to the actuator tip, distributed the load along the width of the spar cross-section. For deformation monitoring, fiber optic sensors were installed at various points along the longitudinal axis. These sensors were placed both externally on the upper and lower skin and in the bonding layer between the stringers and skins.

Standing the adopted configuration, the spar was modeled as a simply supported beam. The boundary conditions of the flexural test replicated the actual constraints used in the laboratory test. For installation reasons, due to the complexity of the implemented overall sensor networks, and other instruments, the spar end terminations were not symmetrical. This was considered in the model: the first row of support constraints was placed 32 mm from the end section close to the D3 damage region, while the other row of support constraints was placed at 56 mm from the other end. The acting load was simulated as concentrated forces deployed along a single row at the midspan of the spar, since the cylindrical termination of the actuator induced a line-distributed force, [19,20,21]. An image of the experimental beam is reported in Figure 8.

## 3. The SHM Methodology Outcomes Validation

The algorithm herein presented is based on the Local High-Edge Onset detection [1,19]. It tries to consider structural damage as an edge discontinuity along the strain energy signal. Edge onset can be tracked in both the space and time domains’ correlating values of adjacent sensors. The flow-chart of the methodology is presented in Figure 9. In the following, a sample outcome of the main post-processing phases, from data preparation to data clustering, is reported.

The cross-correlation function is the core of this process as it represents the measure of similarity of two signals, which, in this case, is applied both as a function of a time shift of each single sensor and applied as an iterative spatial translation of current-to-next sensors at a fixed time.

The sensor step (or window lag) must be chosen according to the minimum damage length to be detected (at least three sensing points as the sample rate). This length is a design spec for the SHM system and it is tailored to the specific application. In this case, the structural analysis and safety factors provided a critical de-bonding extension of 80 mm [20,21]. According to this design input, the “window lag” of 5 mm provided by a distributed fiber is compatible with the minimum damage dimension to be detected. In particular, the damage layout introduces three damages of different extensions (80 mm, 70 mm and 40 mm) placed in an unknown position in order to verify the detection capability of such an SHM system in also detecting the damage of the lower length (Figure 10).

It must be noted that the embedded fiber optic length is 1600 mm but the total length of the sensor is about 2000 mm, so the exceeding length has not been included in the post-processing analysis. As sketched in Figure 10, the extra length of the fiber has been used as a stand off cable from the egress point (left side of the picture 11) and routed until reaching the optical interrogator. The embedded segment of the sensor is highlighted by the blue window in Figure 11 and Figure 12.

By considering a data streaming set provided by a time acquisition (a few seconds at a 10 Hz sample rate) of distributed strain sensors (Figure 11) as input signals, the cross-correlation function can be written using Equation (1) as follows:(1)$$R_{ij}(k)=\frac{1}{N}\sum_{l=0}^{N-1}\varepsilon_{i}\left(k\right)\varepsilon_{j}\left(k+∆k\right)$$

If the strain at the current acquisition (*i*) is not affected by any variation with respect to time (∆k) or space lag (*j*), and similarly, when the signals at different sensor location points coincide, the value of Equation (1) is maximized and corresponds to the auto-correlation (energy density). As the goal is to look for a change in the structural stiffness, both strategies (time similarity or space location similarity) can be adopted, and the autocorrelation function can be used as the reference signal. The outcomes of Equation (1) are provided as examples in Figure 12, where the values are expressed as adimensional units (A.U.s). In addition, the mean value of the outcome data set is estimated and reported as a threshold limit (TL) (blue line) used to filter eligible sensors. Readouts below the TL are discarded, while readouts higher than the TL are kept.

By setting the auto-correlation function of the current responses as a vector,
(2)$$R_{max}\left(k\right)=[max(R_{ii}\left(K\right))]$$
where i=1,2,...,n is the response from the current measurement sensor at pointer *i*; the relative change in the cross-correlation function with respect to the reference auto-correlation vector is defined as a damage index as follows:(3)$${CDI}_{i}=([(R_{ij})]-[R_{max}])/[R_{max}]$$

In the absence of a jump/edge, the damage index (3) will be small. On the contrary, if an edge is present, then the two function values in (3) will be quite different. Once the cumulative damage index (CDI) is estimated by Equation (3) for each sensor, a selection of eligible data sets with the highest probability of occurrences and persistency is provided by using a level of confidence by calibration of the TL. The calibration is a sort of “algorithm tuning” and it can be carried out considering whatever reference condition of the current structure under representative load. Of course, the current status calibration does not take into account possible damage already present or possible stiffener variations due to design and hence prone to be considered as false positives (Figure 13). The figure describes, with yellow bars, the sensors corresponding to the real damage positions, and the green bars correspond to the SHM output having a higher strain gradient. In this case, this configuration is considered as a baseline reference. Since the damage areas are already presented (artificial damage deposited during manufacturing process) a baseline configuration was acquired by using fiber optic on the opposite spar cap in a healthy condition.

In this case, Figure 14 corresponds to the SHM output provided by using the strain coming from the fiber optic monitoring of the damaged cap. As before, the black bars correspond to each sensor of the fibers, with the damage positions in yellow and the sensors indicating the damage positions in red.

## 4. The POD Methodology

The conventional POD assessment procedure is described in [22]. Just for sake of completeness, the popular definition states that “the standard tolerance limit assuming a normal distribution” is determined so that one can state with (1-α)% confidence that at least Φ% of the data fall within the given limits. This is commonly stated as something like “a 95% confidence interval for 90% coverage”. In order to assess the POD_90/95_ of the SHM system, according to [23], the method should have to consider all cases the SHM system might encounter, such as different environmental conditions, different structures, and so on.

Because the collected data at our disposal are limited due to the fixed configuration of the structural component under testing, it is not possible to apply the proposed assessment procedure. For this reason, the threshold value is estimated by a similar analysis procedure. The main difference is to consider a B-basis one-side tolerance limit test, by adopting a k_B_ value that is numerically tabulated for the number of data at our disposal [23]. An example of this procedure is provided in Figure 15 where data from a set of damaged coupons are analyzed [1]. This figure represents, in a radar way, the distribution of detected damage certified by a c-scan procedure (dotted curve) compared to the damage detection by an SHM system (straight curve). The deviation of the two distributions corresponding to a 95% confidence interval for 90% coverage is estimated by adopting a predefined k_B_ value in Equation (4).

The applied procedure is described as follows:

First, numerical data collection is carried out based on a validated numerical model. Basically, tests for data collection are carried out taking into consideration the current and fixed damaged configuration of the spar. Herein, three de-bonding areas realized in a controlled way during manufacturing are considered as reference conditions for the POD test.

Second, the correlation between the output of the SHM algorithm and the de-bonding lengths is found by considering different loading conditions and constraints necessary to create a sufficient data set for analysis.

Third, threshold values are theoretically determined for each one of the de-bonding lengths using Equation (4) in the hypothesis of normal distribution. In Equation (4), *b_th_* is the threshold value, μ and *σ* are, respectively, the average and standard deviation of the estimated data, and *k_B_* corresponds to the B-based one-side tolerance limit, numerically tabulated [22] according to the available number of data.
(4)$$b_{th}=\mu \mp k_{B}\sigma$$

In order to collect the data to assess the POD of the SHM system, a static flexural analysis that considers two supports is established at a certain, fixed position, as well as a running vertical concentrated load covering all the spanwise length of the beam with an 8 mm step. The load position starts at the first support (56 mm) and ends at the second support (1568 mm), since external locations would have no meaning in this analysis (non-equilibrated load-reaction system, leading to rigid motion). In this way, 187 static flexural responses are obtained, and an almost comprehensive analysis is performed, i.e., retrieving all the possible and sometimes very different structural responses of the beam under a concentrated load (Figure 16), within the limits of the adopted mesh.

As already mentioned, a larger number of tests would have been possible, by referring to the region with a finer mesh; however, it is thought that there would have been no significant additional info in considering the nodes between the 8 mm interval, for the region corresponding to the damage locations. Several strain acquisition points are then defined. Specifically, four optical fibers are chosen and placed ideally within the adhesive layer in the upper skin, where the damage is simulated. Each optical fiber has several sensing points, spaced 8 mm apart, for a total of 201 acquisition points. The fibers are located 4 mm apart from each other, in the width direction, to cover the entire width of the cap (edges are excluded). In Figure 17, the four fibers are highlighted as four green lines, while the yellow dots indicate the sensitive points (strain acquisition points) of the fibers.

The process of performing 187 static runs is automated by a MATLAB 2021a routine, interfacing with the MSC NASTRAN 2021 code. The running positions of the concentrated force (F1, F2, ..., F187) are reported in the *include* file. Each run takes about 4 min to be performed. At the end of the process, a txt file is compiled for each run, which contains four lines where the strain information is reported. Each line corresponds to a single virtual fiber; each point of the generic line contains a strain value (associated with the position, indicated as a yellow dot in Figure 17 above).

The input to the SHM algorithm is represented by 187 files containing selected strain values of the spar for each point of application of the load. Each of those files is a matrix of four rows and 201 columns; the rows are representative of distributed sensors, in a very dense configuration; the column represents instead the number of the sensors deployed along each of the four fibers. As was written before, the difference between the load positions (187) and the sensors (grids) number (201) is due to the fact that the load positions do not consider the regions external to the supports. The finite element (FE) mesh is not uniform, while the load is moved at regular intervals of 8 mm steps.

The output is made of a single file, where numbers are collected into a wide matrix, made of 187 lines by 201 columns (in other words, each run provides a single output line). The elements contain a Boolean value (1 or 0) for each point, which is set depending on the identification of a high edge onset (1) or not (0). Such a matrix is then elaborated for retrieving information about the capability of detection of the proposed algorithm, evaluated over the whole range of possible load positions.

By applying Equation (4), the mean (green circle) and standard deviation of the estimated data are plotted as dotted circles where the upper (blue) and lower (yellow) bounds are provided by using the B-based one-side tolerance limit, numerically tabulated for 187 data points (Figure 18). The deviation of the two distributions corresponding to a 95% confidence interval for 90% coverage is estimated by adopting a predefined k_B_ value in Equation (4). This figure represents the distribution of detected damage certified by a c-scan procedure (red sector) compared to the damage detection by an SHM system (black curve).

## 5. Results Analysis

The output file is processed in MATLAB 2021a to perform an exhaustive analysis of the results with respect to the specific load, herein considered, moving all over the beam span, leading to a preliminary POD estimation. Within the limits of the hypotheses above (comprehensive analysis over a stepped-moving load), such an investigation establishes the exact probability the Structural Health Monitoring (SHM) code has to identify the damage when the reference structure is subjected to a point load, for the assigned damage system. Two types of outcomes are explored, of global and local nature, respectively.

Globally, the Boolean values of the output matrix are summed by each row. The output then expresses the number of occurrences that the SHM code predicts a certain node to be associated with the existence of a flaw. A visual representation of that result is reported in Figure 19. The 201 virtual sensor IDs are reported along the *x*-axis, while the number of positive occurrences for each sensor is reported along the *y*-axis. In this two-dimensional representation (irrespective of the width), the red bands represent the extension of the three damage areas we refer to, the green lines represent the lines of discontinuity associated with the skin’s change in thickness, and the yellow lines represent the point supports. The number of occurrences is then normalized by the number of global runs (187, since the load path is limited to occur between the supports), so to establish a number which is the probability the SHM code has to identify the presence of damage concerning the overall considered cases). In particular, the probability indicates an event where the point is the edge of a damage area.

Observing the height of the black bars, the expression of the response of the sensors, it can be seen that the highest values of the function are concentrated in the damage edge areas, as expected, indicating optimal performance of the algorithm. Such a probability always overcomes 70% for all six edges and overcomes the threshold of 90% in two out of the six cases. Four of the seven thickness tapering lines (one of the eight is incorporated within a damaged region) are also detected. It is also relevant to see how one of the support lines is identified, while not the other. A support should lead to a nominal zero strain, so it should be difficult to detect that issue. It is a matter of further investigation to understand the behavior of the SHM code in the presence of such singularities. It should be added that, since the tapering and the constraint areas are well-known from the beginning (information from the design), the related output may be easily discarded. On the other hand, a problem arises concerning the possibility the fault edge is located at the boundaries of the tapering regions; this point also deserves further attention.

A further analysis is carried out by focusing on the local behavior of the SHM algorithm, specifically referring to the sensors within the damaged area domain, herein defined as an extension equal to twice the length of the single flaw. In this way, for the damage D3, D4, and D2, lengths of 104, 192, and 160 mm were considered, respectively. From the output matrix, the columns are extracted, corresponding to sensors whose position falls in those domains. As before, the direct sum of the Boolean values is performed, but this time along the lines, instead of along the columns; the output number now represents how many sensors within the fault domains are indicated by the algorithm as indicative of a possible irregularity, for each run. The result is reported in Figure 20. The *x*-axis is this time referred to as the load case ID (1–187).

The first relevant outcome is that in no case does the number of highlighted sensors fall to 0; in other words, there is always a sensor in the damage area that is elected by the SHM algorithm as a possible indicator of the presence of damage. Moreover, the cases where only one sensor is identified are extremely few with respect to the other circumstances. Another common behavior is that the most prominent detections (when a large number of sensors in the target region are detected) occur when the exciting force falls within the damage area; since the energy transferred to that particular area is larger as the load is acting there (max strain values, for the considered load), this result seems reasonable.

In detail, the graph on the left of Figure 20 refers to damage D3; in that case, the function appears pretty flat, with most of the values ranging between 2 and 3, with a peak at 7 and a minimum value at 1 in a couple of circumstances. Similar behavior is revealed for damage D2; this time, the function shows flat behavior in the range between 3 and 4, with a peak of 10 as the load moves through its area (the number of investigated sensors is larger, as the damage is wider). Finally, for damage D5, a sort of flat function is detected, placed at 3; the oscillations are less relevant than in the other cases, with a peak of 5 as the load transits along its area.

To envisage the outcomes, independent of the damage extension, and therefore on the considered sensors, it is useful to perform a normalization of the values with respect to the maximum number of considered sensors for each damage area. In other words, the absolute number of detected sensors is divided by the number of sensors laying in the defined damage domain, shown in Figure 21. The maximum value of such a percentage or figure of merit is achieved for flaw D3, with peaks approaching 50% in the best case, while most of the data range between 15 and 20%. For flaw D2, the peak value reduces to 40%, while the most of data are located in a range between 12 and 15%. In the case of flaw D5, the max feature falls to 24%, while the most common outcomes are around 14%. Such values may appear relatively low, but they should be evaluated with respect to the specific peculiarity of the implemented algorithm, which is focused on detecting the edges of the damage and not the entire extension of the fault. Since the damage regions are not known in real world cases, it would also be interesting to compare the outcomes for the sensors in positions with no damage (see Figure 20b and Figure 21b).

## 6. Conclusions

The activities of the POD analysis have moved from the FEM previously validated by experimental tests on a composite full-scale wing box spar. The SHM algorithm that was tested was already applied and verified on the real test article, in addition to its numerical representation (the same herein implemented), providing good damage detection capability. In this scenario, the conducted analysis represents a first approach to the estimation of the robustness and sensitivity of the proposed SHM system. A single item was studied, under a specific load configuration, applied exhaustively along its possible positions, i.e., the whole length of the spar.

In this case, damage edges are detected pretty often, as the associated sensors show a very high frequency of TRUE occurrences. In addition, it can be noted that some extra positives are also detected, indicating structural stiffness variation linked to local imperfections, stiffening, or other architectural discontinuities. As part of the design process, these items have been known and assessed since the very beginning and do not represent a shortfall; they can be logged, tracked, and then eventually filtered by using the output as an offset for the next evaluations.

Since this methodology is based on relative comparisons and correlations, it is able to provide very local information (place, size), independently by the excitation source. In turn, it does, however, mean that the output is affected by the way the elastic energy is distributed within the structure (strain map), which may depend on the load type but also on how it is spread over the item (position, direction, distribution, and so on), as a function of the constraints.

In detail, the analysis allowed us to assess some characteristics of the system that enable us to classify the proposed methodology, even from a quantitative standpoint. For instance, it can be seen that a recursive number in the sensors’ prediction is about 15% of true positives as the device is located within the damage region vs. the analyzed cases. It does mean that at least seven sensors should be deployed in the damage region so that the fault is always detected. In turn, it also means that at least seven analyses shall be conducted to achieve that result. Of course, this is something related to the specific test case, herein analyzed, and many other investigations are necessary before a final statement shall be assessed. A theoretical model could be way more powerful, but it would require time and uncertain outcomes. That is why it is more viable to envisage a wider numerical campaign in the future, based on the same architecture. A further step would be then to move to other specimens, staying in the beams’ domain, and other kinds of test articles (for instance, plates and stiffened plates), to expand the knowledge on the matter. Also of some interest is a complement to the analysis regarding sensors that are external to the damage region, which show a probability of about 2% of being detected as a possible area of irregularity. Combining the previous and the current result, this does mean that the “background noise” of the system is meant as the value that should be assumed by each sensor as a false positive amount at one magnitude less that the actual detection. Practically, this does mean that the actual probability of distinguishing real damage occurrence falls to 13%, leaving the minimum sensor density unvaried vs. the previous predicted necessities. Finally, it shall be considered that the extremes of the damage areas seem to have a very high probability of being detected, attaining 70% at minimum. This is because the border area is very small by definition, and therefore it gives a constraint to the sensor network density. As a true preliminary outcome, it can be seen that larger fault domain, until having two sensors, experience a high POD, while the shorter ones have single sensors. Then, it could be assumed as a very preliminary guess that, to attain sufficient readability, at least a coverage amounting to about 6% of the damage extension is necessary. As before, this result shall be verified vs. other and more extensive analyses. As can be seen, they are excellent indications of the design of a proper sensor network.

The excellent results achieved in this first study aimed at a POD assessment of the implemented system should be aimed at attaining more general results. As a first further step, it is considered that the same test article should be referred to while also imagining at least two classes of variability: damage size and position, and load variability. The first instance is easier to run, and therefore it is the most suitable to be applied. It is believed that, by always considering a certain region of the beam (perhaps keeping the disbonding area in at the interface between the spar and the top plate), the damage size should be varied. The issue is not so complex after all, since the damage extension is limited (by design) to 80 mm, while its width can only achieve 20 mm (the spar cap width). If the reference mesh is considered, a number of 5 × 10 possible damaged regions shall be considered, a number that should be further multiplied by the possible locations (a number between 170 and 187, depending on the flaw length). Each of the case should consider a different position of the external load (187, again). Above all, the impressive number of more than 1,600,000 cases should be run—something that is not suitable for a reasonable analysis—if it is recalled that the load type is defined. Thus, it is mandatory to approach the issue not as a comprehensive investigation but as a real probabilistic study. In this sense, the variability of the load should be considered, together with some considerations about the constraints (not just the position but the type as well). Moving to other structural components should represent the final step of this first round.

Such an analysis is not trivial and could require a good amount of time and effort. It is therefore recommendable that it would take place after the Technology Readiness Level (TRL) approaches 6 and that the methodology and technology implemented have proven their effectiveness. Since then, studies devoted to exploring the behavior of the SHM code could be still run, focused on specific configurations, time by time. The following schematic may then be drafted for the further activities in this segment:
-Investigating further specific configurations of interests, according to the inputs received from the industry and the final customers.-Targeting the attainment of TRL 6 (or close to it) for the proposed technology.-Performing a probabilistic analysis on a specific test article, having the following variables as reference, for instance (not an exhaustive list):Damage size;Damage location;Constraint type (location variability is not considered a relevant parameter in a first step);Load type and location.-Performing a probabilistic analysis by referring to various relevant test articles, as indicated by the industry and the final customers.

## Figures and Tables

**Figure 1 sensors-24-05216-f001:**
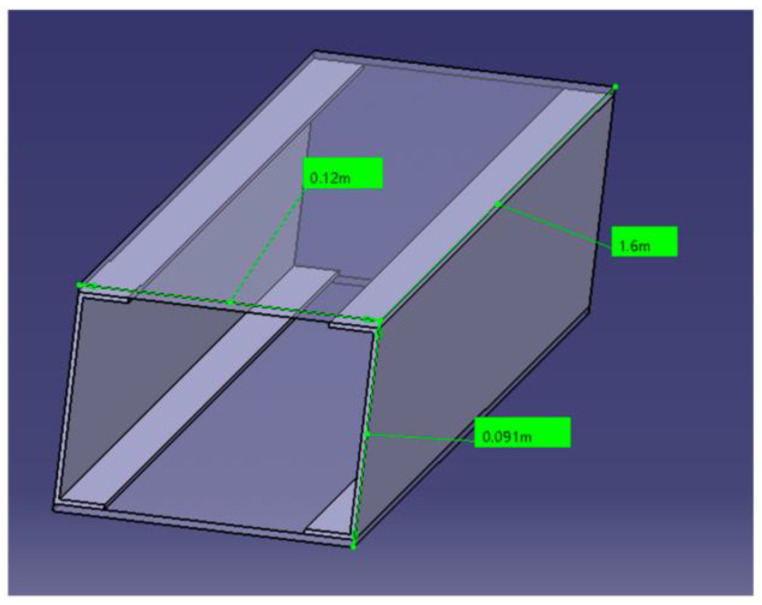
Schematic view of the spar used for the POD activities.

**Figure 2 sensors-24-05216-f002:**
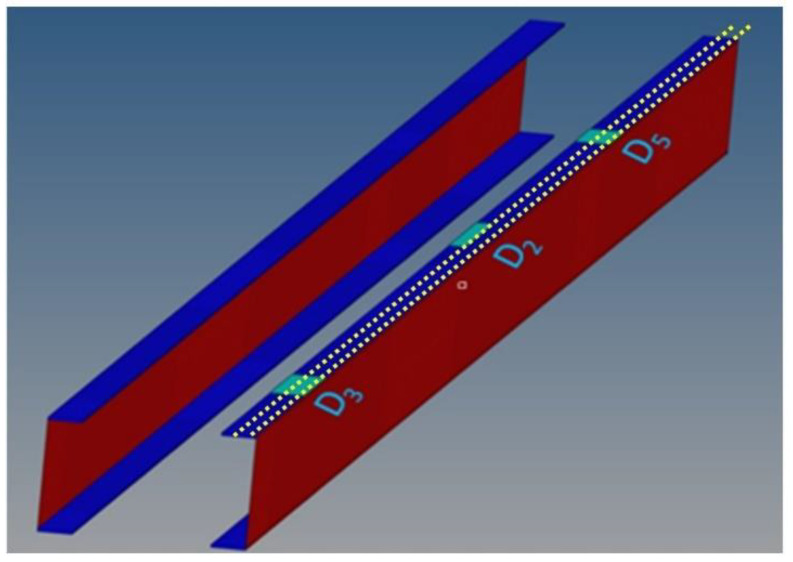
Detail of the C-spars modeling, with the position of the imposed damage regions highlighted and the fiber optics (dotted lines). Upper and bottom skin panels are not visualized. Caps are depicted in blue, while the webs are plotted in red.

**Figure 3 sensors-24-05216-f003:**
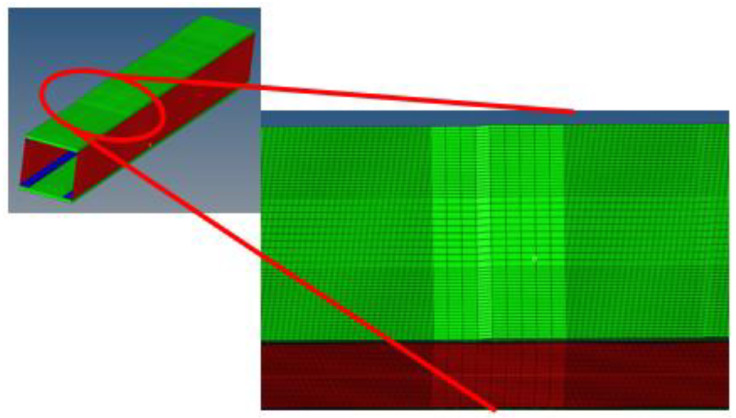
Mesh refinement in proximity to the damaged areas.

**Figure 4 sensors-24-05216-f004:**
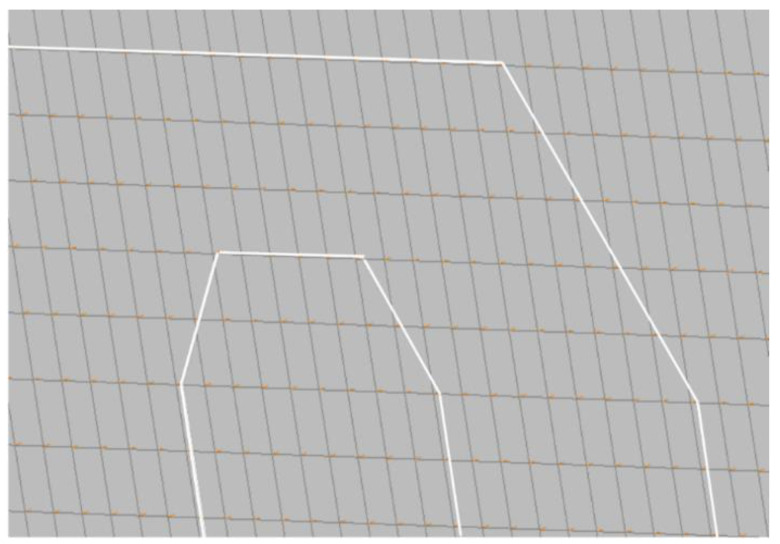
Application of the Segment method for modeling optical fibers (white line): detail of the curved path.

**Figure 5 sensors-24-05216-f005:**
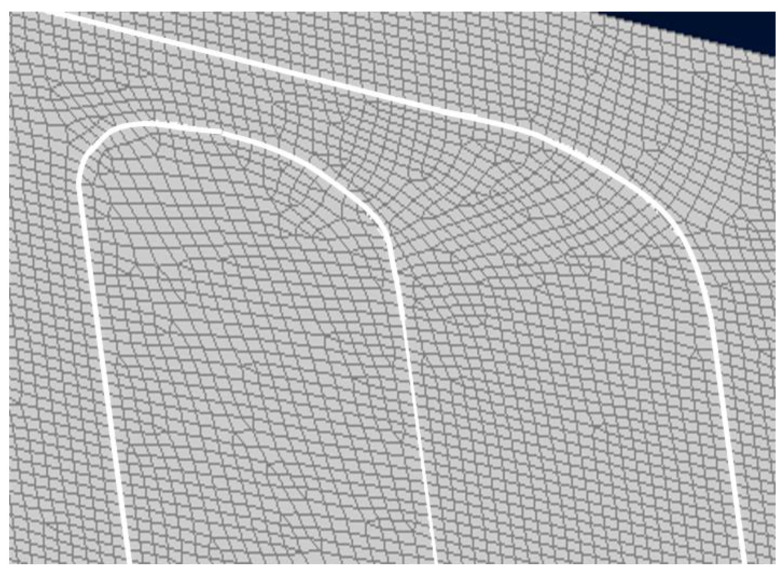
Application of the Merge method for modeling optical fibers (white line): detail of the curved path.

**Figure 6 sensors-24-05216-f006:**
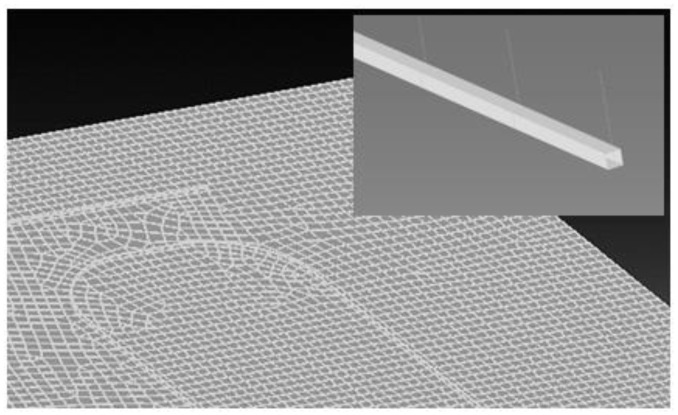
Application of Virtual Contact method for modeling optical fibers (white line): a detail of the curved path.

**Figure 7 sensors-24-05216-f007:**
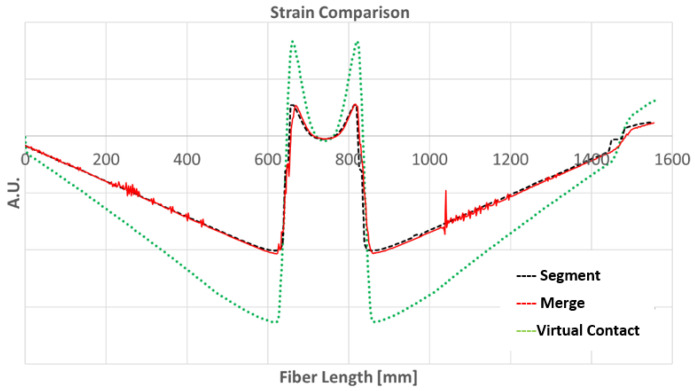
Finite element simulation of fiber optic strain measures by *Segment*, *Merge,* and *Virtual Contact* methods.

**Figure 8 sensors-24-05216-f008:**
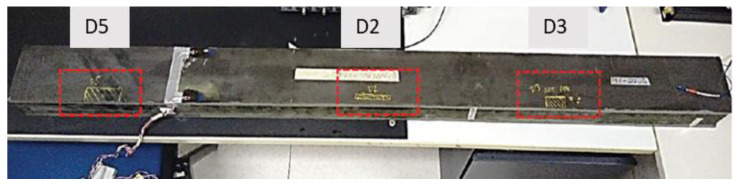
Aeronautical full-scale composite spar.

**Figure 9 sensors-24-05216-f009:**
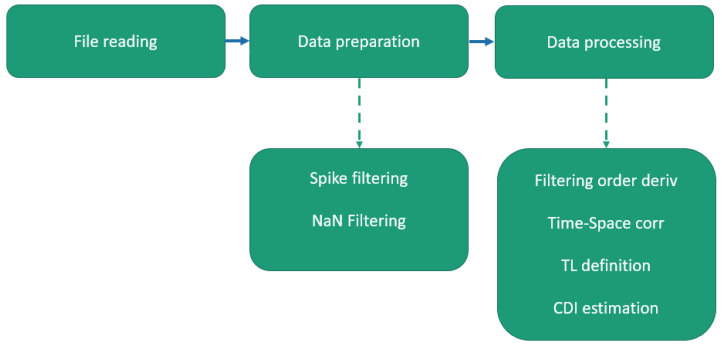
SHM methodology flow-chart based on cross-correlation analysis.

**Figure 10 sensors-24-05216-f010:**
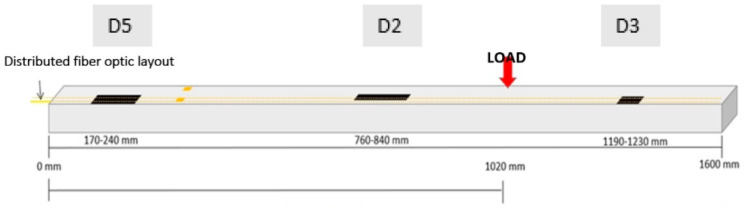
Simplified sketch of the beam with position of damage (black rectangles), fiber optics (yellow lines), and load position (red arrow).

**Figure 11 sensors-24-05216-f011:**
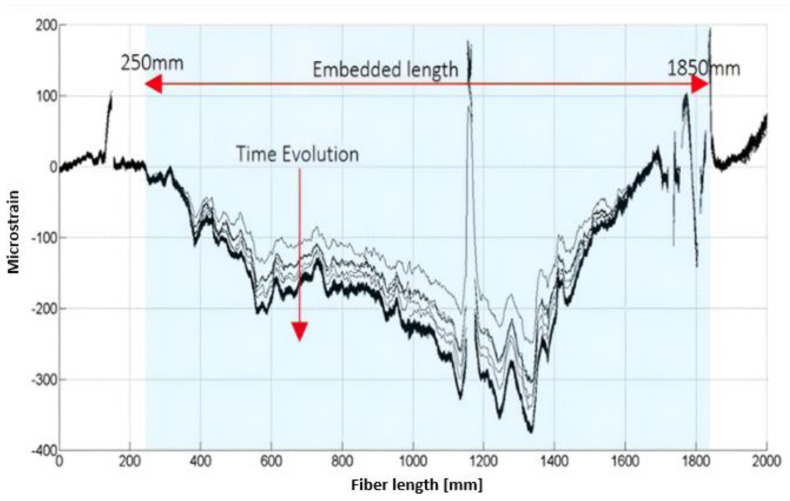
Experimental strain map during quasi-static loading.

**Figure 12 sensors-24-05216-f012:**
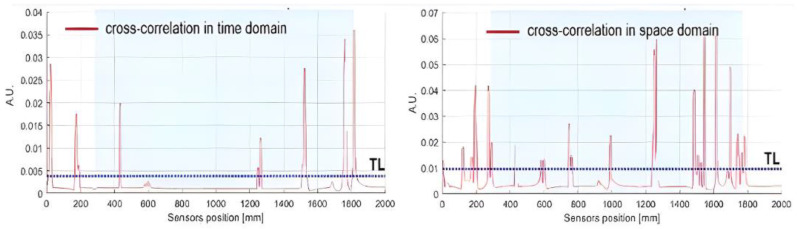
SHM feature extraction: time domain cross-correlation (**left**); space domain cross-correlation (**right**). Threshold limit (TL) (blue line).

**Figure 13 sensors-24-05216-f013:**
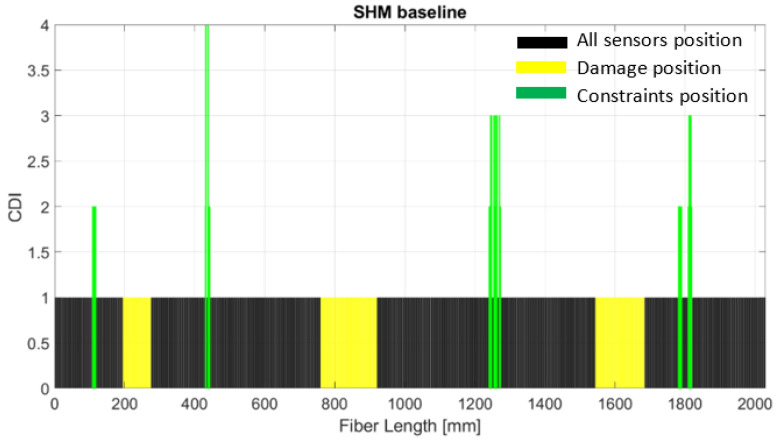
SHM readout of the baseline structure for the healthy spar cap.

**Figure 14 sensors-24-05216-f014:**
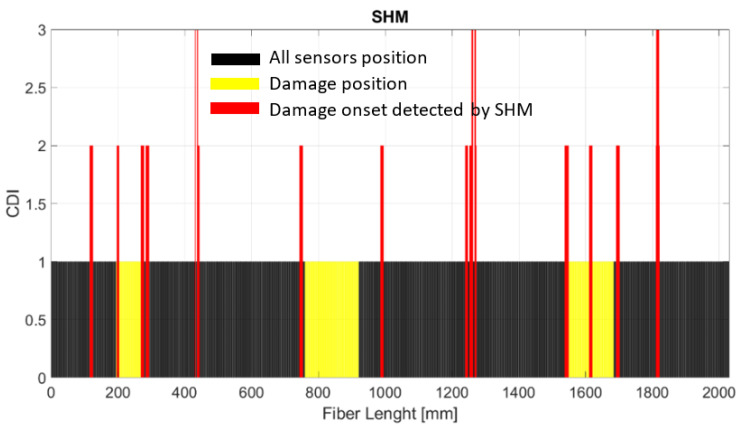
SHM readout of the damaged structure.

**Figure 15 sensors-24-05216-f015:**
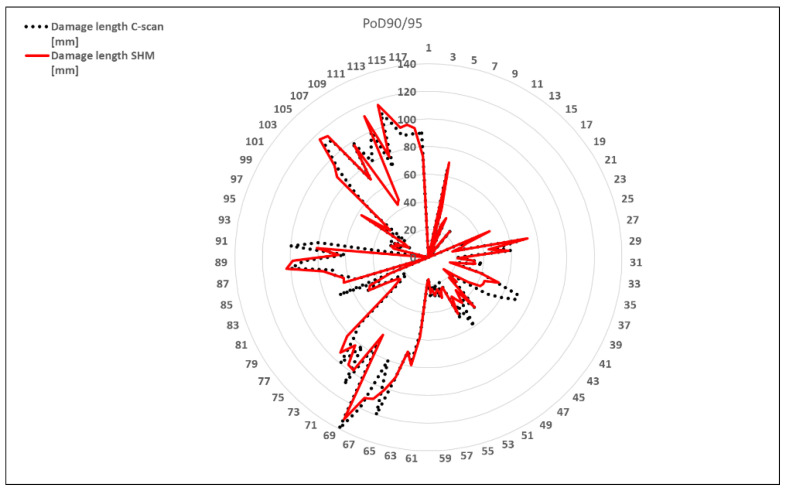
Example of the relation between measured response by SHM and de-bonding length by C-scan.

**Figure 16 sensors-24-05216-f016:**
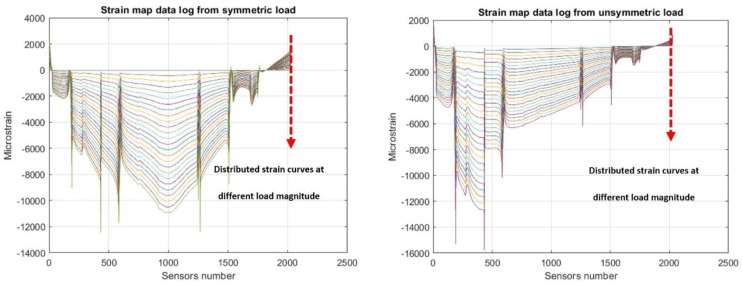
Examples of numerical estimation of strain responses by different loading position: symmetric loading (**left**); asymmetric loading (**right**).

**Figure 17 sensors-24-05216-f017:**
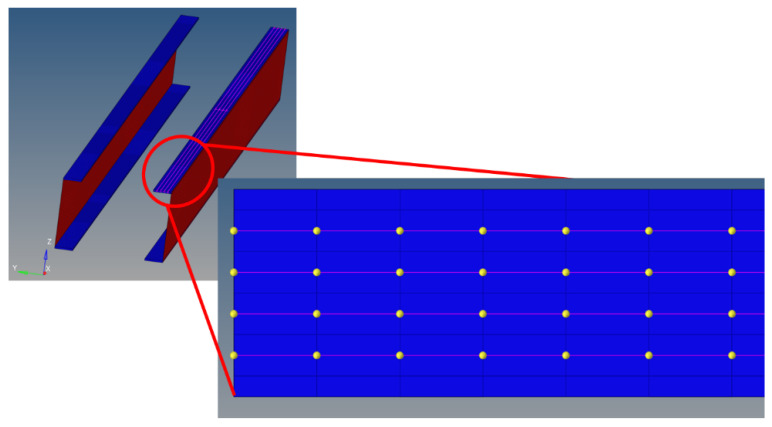
Detail of the composite beam. The top skin has been removed to provide a better view of the 4 fibers, embedded at the interface between the adhesive layer and the top skin. The 4 installed fibers are represented by 4 fine lines, while the yellow dots indicate the sensitive points, where the strain values are retrieved.

**Figure 18 sensors-24-05216-f018:**
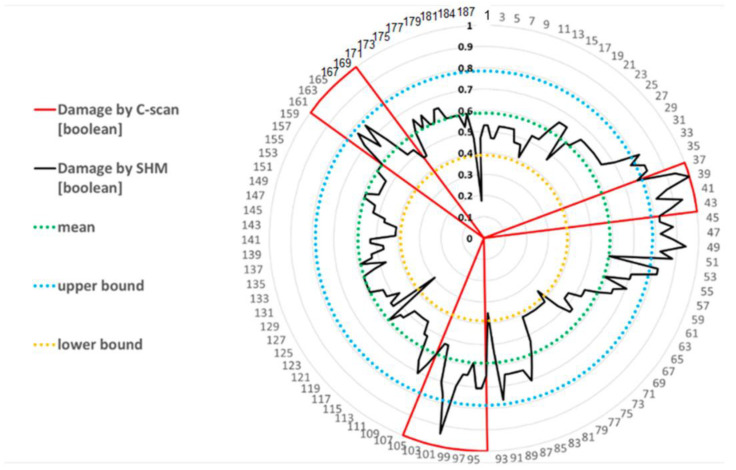
Relation between measured response by SHM (black distribution) and real de-bonding length (red distribution). Relation with threshold values by B-basis one-side limit (yellow and blue dotted circles) by using k_B_ = 1.456 numerically tabulated for a data set of 187 elements, according to one-side tolerance limit of the normal distribution.

**Figure 19 sensors-24-05216-f019:**
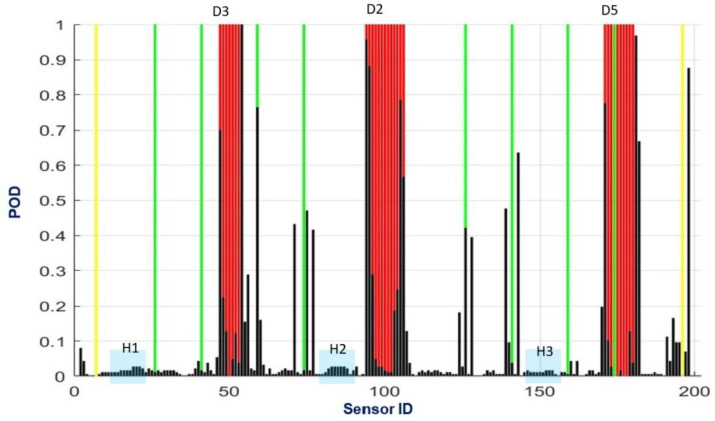
Outcome of the SHM algorithm, in terms of the probability of detecting the damage edges, for the considered configuration (given constraints and damage areas; running point vertical load). The *x*-axis shows the 201 sensor IDs, arranged along the fiber with a constant 8 mm step, while the *y*-axis shows the number of occurrences, normalized with respect to the performed runs, which refer to the damage edge detection. The red bands represent the extension of the three damage areas; the green lines represent the tapering lines (thickness variations); and the yellow lines represent the location of the two supports. The blue rectangles indicate three arbitrary regions of structural healthy conditions.

**Figure 20 sensors-24-05216-f020:**
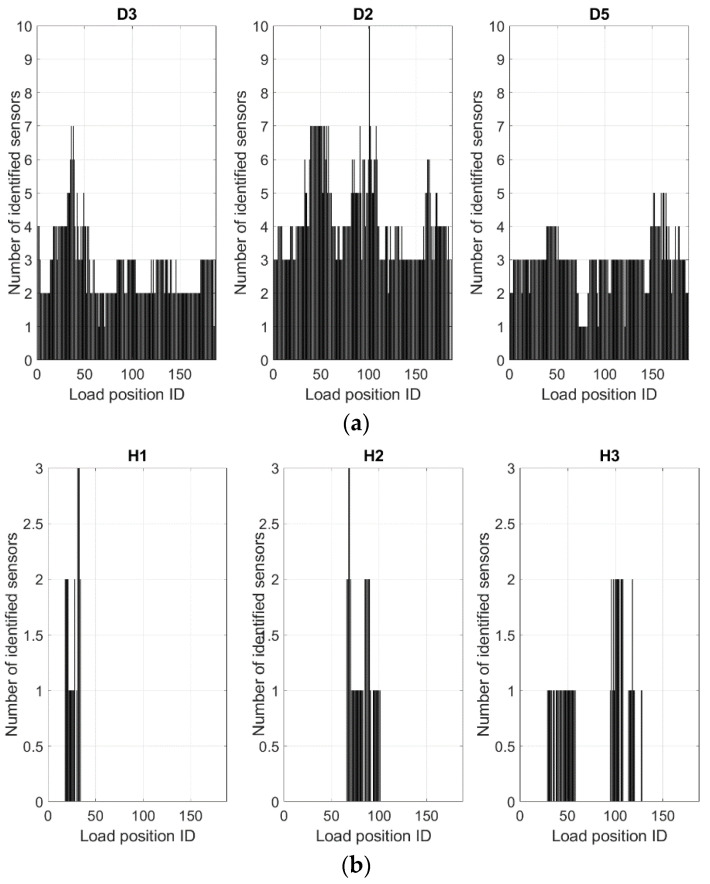
Number of identified sensors: (**a**) the number of sensors identified by the SHM algorithm as indicators of damage occurrence, for each of the 3 damage zones; (**b**) the number of sensors identified by the SHM algorithm for the 3 healthy zones (see Figure 19). REMARK: Top and bottom vertical scales are different.

**Figure 21 sensors-24-05216-f021:**
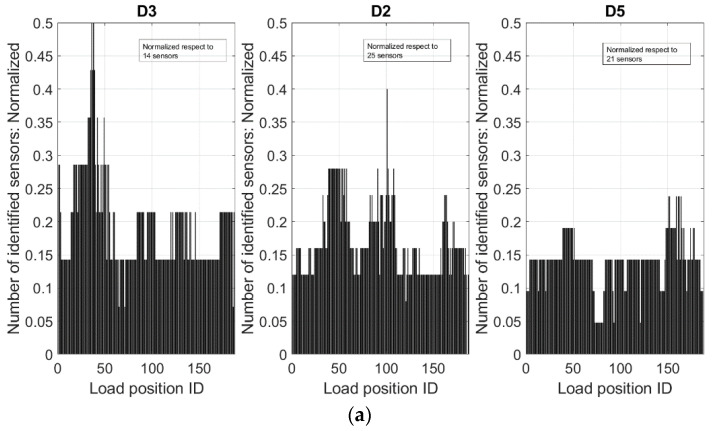
Normalization of the values reported in Figure 20, with respect to the total number of considered sensors in detail: (**a**) normalization by 14 sensors (D3), 25 sensors (D2), and 21 sensors (D5); (**b**) 3 arbitrary healthy zones normalized by 15 sensors. REMARK: Top and bottom vertical scales are different.

## Data Availability

Data are not available publicly for confidentiality reasons.

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
