# Peer review of "Performance Evaluation of Structural Health Monitoring System Applied to Full-Size Composite Wing Spar via Probability of Detection Techniques"

_sensors, 2024, doi:10.3390/s24165216_

Round 1

Reviewer 1 Report

Comments and Suggestions for Authors

This work is a part of ongoing research within the RESUME (Real–Time Structural Health & Usage Monitoring Systems for UAV) project and is devoted to the analysis of the probability of defect detection (POD) to assess the reliability of the SHM system applied to the composite spar of the wing caisson to assess the quality of the connection line. The variability of indicators of artificially created damages in the manufacture of a full-size composite spar was taken as the initial data for a non-exhaustive POD analysis to assess the reliability of the SHM system used to detect defects of the "kissing joint" type. The experiments were carried out on a test product using a finite element model, including the identified defects and a fiber-optic sensor system deployed in the area of interest. The analysis of the collected data for the POD is presented and commented on. Significant results have been obtained in the important field of SHM aviation and space technology. However, these results represent a first approach to assess the reliability and sensitivity of the proposed SHM system. They indicate the directions of further research that the authors are planning.

Comments include a wish to briefly summarize the results of previous research in this project.

Author Response

These authors wish to thank the reviewer for the appreciation of the work. This submitted review includes a summary of previous results in the section of the algorithm and a better description of the methodology and data processing.

Reviewer 2 Report

Comments and Suggestions for Authors

This manuscript utilized POD to finite element model data for structural health monitoring. Since the FE model was validated by experimental results, I think the authors should focus on how POD works on the simulation result. The structure of this manuscript should be improved. As a research article, I expect more information about theoretical and technical analysis. The manuscript at this stage is more like a detailed laboratory report.

(1)Please reduce the quantity of the keywords.

(2)There are lots of compile errors of the references in the manuscript.

(3)The resolution of Figures should be improved.

(4)I am confused about how to create and process the damage regions in Figure 2. Please provide detailed information and some explanation.

(5)It takes 4 pages to explain the numerical model. If the simulation was conducted in a commercial FEM software (please specify the software), only some important steps should be detailed.

(6)Labels of curves should be added in Figure 9 12 13 14 15 17.

(7)Are POD and PoD the same thing? If POD is important for this paper, more theoretical information should be given.

Author Response

Comments 1: Please reduce the quantity of the keywords.

Response 1: Thank you for pointing this out. We agree with this comment. We have reduced the keywords as suggested.

Keywords: Probability of Detection; Structural Health Monitoring; Aeronautic composite structures; De-bonding; FE simulations

Comments 2: There are lots of compile errors of the references in the manuscript.

Response 2: Agree. We have, accordingly, modified the references as you suggest. All the compile errors have been removed.

Comments 3: The resolution of Figures should be improved.

Response 3: Figures resolution improved

Comments 4: I am confused about how to create and process the damage regions in Figure 2. Please provide detailed information and some explanation.

Response 4: The following sentence has been added: This damage zones are modeled simulated the absence of the adhesive at the interface between cap and skin (kissing bond phenomenon), by assigning to the elements of the adhesive a negligible Young modulus (10-8 order of magnitude, just to avoid numerical instability). This value represents a significant reduction in stiffness compared to the inherent properties of the undamaged bonding material.

Comments 5: It takes 4 pages to explain the numerical model. If the simulation was conducted in a commercial FEM software (please specify the software), only some important steps should be detailed.

Response 5: As suggested by the reviewer the paragraph has been reduced holding, as much possible, trying to hold important steps details

Comments 6: Labels of curves should be added in Figure 9 12 13 14 15 17.

Response 6: Labels have been added

Comments 7: Are POD and PoD the same thing? If POD is important for this paper, more theoretical information should be given.

Response 7:Yes, POD and PoD have the same meaning but as suggested by the reviewer to adopt POD for all. It was a TYPO.

The main motivation for a POD analysis is better explained in the abstract, introduction and conclusion. (lines 30-32) (lines 536- 561).

Reviewer 3 Report

Comments and Suggestions for Authors

The paper under review deals with a numerical procedure to asses damage/defect identification and localization on composite wing spar. It is opinion of the present reviewer that the paper could be considered for publication after the following major revisions will be addressed:

1. In the Introduction and Conclusions it is suggested to explain with adequate details the novelty of the paper also with respect to previous papers from same authors, particularly the one cited in reference [20];

2. in Section 2, at page 7, it is suggested to add one or more figures/sketches of the model spar with geometrical specifications of the deployed sensors;

3. In Section 3, at line 221, the Local High-Edge Onset detection algorithm is mentioned without any reference;

4. The sentence at lines 234-236 is not clear in the present form. It is suggested to specify the meaning of "window lag" for example by adding more details in Figure 11. Is is also worth to be explained how it could be designed according to the damage dimensions to be detected since this information is usually not known;

5. In Figure 12 , several strain plots are reported. Please clarify what are the several plots referred to? Please add Legend or explain in the text;

6. Why does the fiber length vary from 0 to 2000mm in Figures 12-13-14-15-17  but the beam length is equal to 1600mm, as reported in Figure 11?;

7. the sentence "A threshold limit (TL)." at line 272 is isolated. Consider to incorporate to previous or following sentences;

8. Consider to comment Figure 13 in the text more accurately by pointing out what "A.U." stands for and what is the meaning of the horizontal dotted blue line;

9. In Figures 14 and 15 the y-axis label indicates "CDI" but it is not defined in the text.;

10. Figure 16 and 19 need to be better explained in the text;

11.  Explain why Kb=1.456 in Figure 19;

12. Since the damage regions are not known in real world cases, it would be interesting to compare the outcomes, discussed in Section 5 and depicted in Figures 21-22 for the sensors in damaged regions, also for the sensors in positions with no damages;

11. The Conclusions mainly address future development of the research on the same topic. It is opinion of the present reviewer that more comments on the novelties and peculiarities of the proposed damage identification method have to be added instead of discussing so extensively on the future development. Moreover, some of the listed future developments could be addressed in a revised version of the present paper to improve it.  

Comments on the Quality of English Language

English Language needs moderate editing.

Author Response

Comments 1: In the Introduction and Conclusions it is suggested to explain with adequate details the novelty of the paper also with respect to previous papers from same authors, particularly the one cited in reference [20]

Response 1: “The results of this study eventually aim at improving the current strategies adopted for SHM for bonding analysis by identifying the intimate behavior of the system assessed at the date.”(lines 30-32)

In detail, the analysis allowed to assess some characteristic of the system that enable to classify the proposed methodology, even from a quantitative standpoint. For instance, it can be seen that a recursive number in the sensors’ prediction is about 15% of true positives as the device is located within the damage region vs the analysed cases. It does mean that at least 7 sensors should eb deployed in the damage region in order the fault is always detected. In turn, it also means that at least 7 analyses shall be conducted to achieve that result. Of course, this is something related to the specific test case, herein analysed, and many other investigations are necessary before a final statement shall be assessed, A theoretical model could be way more powerful, but it would require time and uncertain outcomes. That’s way it’s more viable to envisage a wider numerical campaign in the future, based on the same architecture. A further step would be then t move to other specimens, but standing in the beams’ domain, and other kind of test articles (for instance, plate and stiffened plate), to expand the knowledge on the matter. Of some interest is also the complement to the analysis sensors that are external to the damage region show a probability of about 2% of being detected as a possible area of irregularity Combining the previous and the current result, it does mean that the “background noise” of the system, meant as the value that should be assumed by each sensor as a false positive amount at one magnitude less that the actual detection. Practically it does mean that the actual probability of distinguish a real damage occurrence falls down to 13%, leaving the minimum sensors density unvaried vs the previous predicted necessities. Finally, it shall be considered that the extremes of a damage areas seem to have a very high probability to be detected, attaining 70% at minimum. The fact is the border area is very small by definition, and therefore it gives a constraint to the sensor network density. As a true preliminary outcome. it can be seen as larger fault domains give till to two sensors that experience a high POD, while the shorter amount to a single one. Then, it could be assumed as a very preliminary guess that, to attain a sufficient readability, at least a coverage amounting to about 6% of the damage extension is necessary. As before, this result shall be verified vs other and more extensive analyses. As it can be seen, they are excellent indication sin the design of a proper sensor network. (lines (536- 561).

Comments 2: in Section 2, at page 7, it is suggested to add one or more figures/sketches of the model spar with geometrical specifications of the deployed sensors

Response 2: the sketch for the deployed sensors is reproduced in Figure 2 and Figure 11

Comments 3: In Section 3, at line 221, the Local High-Edge Onset detection algorithm is mentioned without any reference

Response 3: References 1 and 19 have been added

Comments 4: The sentence at lines 234-236 is not clear in the present form. It is suggested to specify the meaning of "window lag" for example by adding more details in Figure 11. Is is also worth to be explained how it could be designed according to the damage dimensions to be detected since this information is usually not known

Response 4: The following sentence has been added in the paragraph:

The sensor step (or window lag) must be chosen according to the minimum damage length to be detected (at least 3 sensing point as sample rate). This length is a design specs for the SHM system and it is tailored to the specific application. In this case, the structural analysis and safety factors provided a critical debonding extension of 80mm [20, 21]. According to this design input, the “window lag” of 5mm provided by a distributed fiber is compliant to the minimum damage dimension to be detected. In particular the damage layout introduced three damage of different extensions (80mm, 70mm and 40mm) placed in unknown position, in order to verify the detection capability of such a SHM system in detecting also damage of lower length (Figure 11).

Comments 5: In Figure 12 , several strain plots are reported. Please clarify what are the several plots referred to? Please add Legend or explain in the text

Response 5: a legend is added to the figure as suggested.

In addition the sentence in the text as also been added: …By considering a data streaming set provided by a time acquisition (few seconds at a 10Hz sample rate) of distributed strain sensors (Figure 12) as input signals…,

Comments 6: Why does the fiber length vary from 0 to 2000mm in Figures 12-13-14-15-17  but the beam length is equal to 1600mm, as reported in Figure 11?

Response 6: the following sentence has been introduced in the paragraph:

It must be noted that the embedded fiber optic length is 1600mm but the total length of the sensor is about 2000mm, so exceeding length has not been included in the post-processing analysis. As sketched in the Figure 11 the extra length of the fiber has been used as a stand off cable from the egress point (left side of the picture 11) and routed till reaching the optical interrogator. The embedded segment of the sensor is highlighted by blue window in figures 12 and 13.

Comments 7: the sentence "A threshold limit (TL)." at line 272 is isolated. Consider to incorporate to previous or following sentences

Response 7: The sentence has been clarified and aligned to the sentence as suggested

Comments 8: Consider to comment Figure 13 in the text more accurately by pointing out what "A.U." stands for and what is the meaning of the horizontal dotted blue line

Response 8: the figure 13 has been better more commented. The sentences are added to the txt:

The Eq. 1 outcomes are provided as example in (Figure 13) where the values are expressed as adimensional units (A.U.). In addition, a mean value of the outcomes data set is estimated and reported as a threshold limit TL (blue line) used to filter eligible sensors. Readouts below TL are discarded, while readouts higher than TL are kept.

Comments 9: In Figures 14 and 15 the y-axis label indicates "CDI" but it is not defined in the text.

Response 9: the definition of CDI is introduced in the text:

The cumulative damage index CDI is estimated by Eq.3 for each sensor….

Comments 10: Figure 16 and 19 need to be better explained in the text

Response 10: the following sentence has been added within the text:

This figure represents in a radar fashion way the distribution of detected damage certified by a c-scan procedure (dotted curve) compared to the damage detection by an SHM system (straight curve). The deviation of the two distribution corresponding to a 95% confidence interval for 90% coverage is estimated by adopting a predefined kB value in Eq.4.

Comments 11: Explain why Kb=1.456 in Figure 19

Response 11: the sentence is added in the label of figure 19:

…by using kB = 1.456 numerically tabulated for a data set of 187 elements, according to one side tolerance limit of the Normal distribution.

Comments 12: Since the damage regions are not known in real world cases, it would be interesting to compare the outcomes, discussed in Section 5 and depicted in Figures 21-22 for the sensors in damaged regions, also for the sensors in positions with no damages

Response 12: As suggested by the reviewer the outcomes have been provided for 3 arbitrary regions in positions with no damage. Corresponding plots are added in figure 21 and 22.  

Comments 13: The Conclusions mainly address future development of the research on the same topic. It is opinion of the present reviewer that more comments on the novelties and peculiarities of the proposed damage identification method have to be added instead of discussing so extensively on the future development. Moreover, some of the listed future developments could be addressed in a revised version of the present paper to improve it

Response 13: Same details are added in the abstract, introduction and conclusions as follows:

The results of this study eventually aim at improving the current strategies adopted for SHM for bonding analysis by identifying the intimate behavior of the system assessed at the date. (lines 30-32)

This analysis is essential for assessing the actual performance of the structural health monitoring technique, including the sensor network, the algorithm and the preprocessing data analysis. In fact, the performance verification, which is necessarily applied to a case study in order to maintain a direct link with the reality, allows identifying the point the system should be improved on based on the actual outcomes. In order this investigation is satisfactory, the exam of the predictions shall be translated in operational updates and design guidelines. This kind of approach has some degrees of innovation with respect to what can be usually find in literature, and the authors foresee further improvements in handling this strategy in next works. (lines 98- 106)

In detail, the analysis allowed to assess some characteristic of the system that enable to classify the proposed methodology, even from a quantitative standpoint. For instance, it can be seen that a recursive number in the sensors’ prediction is about 15% of true positives as the device is located within the damage region vs the analysed cases. It does mean that at least 7 sensors should eb deployed in the damage region in order the fault is always detected. In turn, it also means that at least 7 analyses shall be conducted to achieve that result. Of course, this is something related to the specific test case, herein analysed, and many other investigations are necessary before a final statement shall be assessed, A theoretical model could be way more powerful, but it would require time and uncertain outcomes. That’s way it’s more viable to envisage a wider numerical campaign in the future, based on the same architecture. A further step would be then t move to other specimens, but standing in the beams’ domain, and other kind of test articles (for instance, plate and stiffened plate), to expand the knowledge on the matter. Of some interest is also the complement to the analysis sensors that are external to the damage region show a probability of about 2% of being detected as a possible area of irregularity Combining the previous and the current result, it does mean that the “background noise” of the system, meant as the value that should be assumed by each sensor as a false positive amount at one magnitude less that the actual detection. Practically it does mean that the actual probability of distinguish a real damage occurrence falls down to 13%, leaving the minimum sensors density unvaried vs the previous predicted necessities. Finally, it shall be considered that the extremes of a damage areas seem to have a very high probability to be detected, attaining 70% at minimum. The fact is the border area is very small by definition, and therefore it gives a constraint to the sensor network density. As a true preliminary outcome. it can be seen as larger fault domains give till to two sensors that experience a high POD, while the shorter amount to a single one. Then, it could be assumed as a very preliminary guess that, to attain a sufficient readability, at least a coverage amounting to about 6% of the damage extension is necessary. As before, this result shall be verified vs other and more extensive analyses. As it can be seen, they are excellent indication sin the design of a proper sensor network.   (lines 532- 557).

Round 2

Reviewer 3 Report

Comments and Suggestions for Authors

The authors have addressed the reviewer's comments satisfactorily, and the paper is now suitable for publication in its current form.

Comments on the Quality of English Language

Please check the text to correct small spelling and typographical errors.